# Programmed Death Ligand 1 Regulatory Crosstalk with Ubiquitination and Deubiquitination: Implications in Cancer Immunotherapy

**DOI:** 10.3390/ijms25052939

**Published:** 2024-03-03

**Authors:** Soon-Bin Kim, Soonjae Hwang, Ji-Young Cha, Ho-Jae Lee

**Affiliations:** 1Department of Health Sciences and Technology, GAIHST, Gachon University, Incheon 21999, Republic of Korea; orot0818@naver.com; 2Department of Biochemistry, Lee Gil Ya Cancer and Diabetes Institute, College of Medicine, Gachon University, Incheon 21999, Republic of Korea; soonjae@gachon.ac.kr

**Keywords:** PD-L1, PTM, ubiquitination, deubiquitination, immunotherapy

## Abstract

Programmed death ligand 1 (PD-L1) plays a pivotal role in cancer immune evasion and is a critical target for cancer immunotherapy. This review focuses on the regulation of PD-L1 through the dynamic processes of ubiquitination and deubiquitination, which are crucial for its stability and function. Here, we explored the intricate mechanisms involving various E3 ubiquitin ligases and deubiquitinating enzymes (DUBs) that modulate PD-L1 expression in cancer cells. Specific ligases are discussed in detail, highlighting their roles in tagging PD-L1 for degradation. Furthermore, we discuss the actions of DUBs that stabilize PD-L1 by removing ubiquitin chains. The interplay of these enzymes not only dictates PD-L1 levels but also influences cancer progression and patient response to immunotherapies. Furthermore, we discuss the therapeutic implications of targeting these regulatory pathways and propose novel strategies to enhance the efficacy of PD-L1/PD-1-based therapies. Our review underscores the complexity of PD-L1 regulation and its significant impact on the tumor microenvironment and immunotherapy outcomes.

## 1. Introduction

The immune system is a complex network of cells and molecules that protects the body against infections and malignancies. Immune checkpoints such as cytotoxic T-lymphocyte-associated antigen-4 (CTLA-4), programmed death 1 (PD-1), programmed death-ligand 1 (PD-L1), lymphocyte activation gene-3 (LAG-3), and T-cell immunoglobulin and mucin-domain containing-3 (TIM-3) are central to maintaining self-tolerance and modulating immune responses [1]. However, cancer cells often use these checkpoints to evade immune surveillance, placing them at the forefront of cancer immunotherapy research [2]. The emergence of immune checkpoint inhibitors as cornerstones in cancer therapy has revolutionized the management of various malignancies. Among these, PD-1 and its ligand PD-L1 axis are pivotal for regulating the immune escape mechanisms of cancer cells [3].

PD-1, a member of the B7-CD28 superfamily, is a 55-kDa transmembrane protein predominantly expressed on activated T cells. In 1992, in a T-cell hybridoma and hematopoietic progenitor cell line in an apoptotic state [4], PD-1 served as a crucial immune checkpoint receptor that modulated the immune response against antigens and maintained the balance between T-cell activation and tolerance [5]. PD-L1 is one of the two ligands of PD-1. Unlike PD-1, which is mainly found in T cells, PD-L1 is expressed in a wide range of cells, including tumor cells, dendritic cells, macrophages, and some non-hematopoietic cells [6]. The interaction between PD-L1 and PD-1 delivers an inhibitory signal that attenuates T-cell proliferation, cytokine production, and the immune response, thus preventing overactivation of the immune system and autoimmunity.

Various cancers manipulate the PD-L1/PD-1 pathway to evade the immune system. Tumors often overexpress PD-L1, which binds to PD-1 on T cells, inducing a state of dysfunction or exhaustion in these cells and allowing for tumor growth and metastasis. This immune evasion mechanism is evident in various cancers, including lung cancer, melanoma, and kidney cancer. The significance of the PD-L1/PD-1 pathway in cancer therapy has been highlighted during the development and success of immune checkpoint inhibitors [7,8]. These therapeutic agents, which target PD-L1 or PD-1, have demonstrated remarkable efficacy in reactivating the immune system against various cancers, leading to durable responses and improved survival rates in several malignancies. This success has solidified the role of the PD-L1/PD-1 blockade as a cornerstone of modern oncologic therapeutics and has spurred extensive research into the mechanisms underlying their function and regulation.

Despite the success of PD-L1/PD-1 inhibitors, challenges such as variable patient responses and the development of resistance remain [9]. Factors influencing these responses include tumor heterogeneity, differential PD-L1 expression, and complexity of the tumor microenvironment. Current research is focused on identifying predictive biomarkers for the response to these therapies and investigating combination strategies to enhance their effectiveness.

Central to the regulation of PD-L1 are post-translational modifications (PTMs), among which ubiquitination and deubiquitination are particularly critical [10,11,12,13]. Various E3 ubiquitin ligases have been identified to ubiquitinate PD-L1, targeting it for proteasomal degradation and thereby influencing the immune escape capabilities of tumor cells. Conversely, deubiquitinating enzymes (DUBs) can stabilize PD-L1 expression and enhance its immunosuppressive function. Understanding this complex regulatory network is fundamental for understanding the biology of tumor immunity and is crucial for identifying novel therapeutic targets within the PD-L1/PD-1 pathway.

The objective of this review was to explore the intricate regulatory crosstalk between PD-L1 and modifications of ubiquitination and deubiquitination. We aimed to provide a comprehensive overview of how these PTMs affect PD-L1 function and regulation and elucidate their implications in the realm of cancer immunotherapy. This study entails a detailed examination of the specific E3 ligases and DUBs involved in PD-L1 regulation, their mechanisms of action, and the potential therapeutic opportunities they present in the ongoing battle against cancer.

## 2. PD-L1 and Its Transcriptional Expression

PD-L1, also known as B7-H1 or CD274, plays a pivotal role in the immunological landscape, particularly in cancer immunotherapy. This transmembrane protein is intricately structured and comprises seven exons located on chromosome 9 (9p24.1). The first exon is non-coding, which sets the stage for the complex arrangement of subsequent exons. Exons 2 through 4 are particularly crucial and encode the immunoglobulin (Ig)-V-like and Ig-C-like domains, which are integral to protein function [4,14]. The transmembrane and intracellular domains are encoded by exons 5 and 6, with the last exon contributing to the intracellular domain and the 3’UTR [6]. PD-L1’s major transcript is 4.2 kbp, alongside a 7.2 kbp transcript and a truncated form lacking the Ig-V-like domain [15]. The full-length protein is ~31 kDa and comprises 290 amino acids, with the extracellular region exhibiting approximately 20% amino acid identity with B7-1, another significant immunological player [6,16].

Tumors evade immune attack and attain self-defense through the adaptive enhancement of PD-L1 expression. This increase is induced by interferon-gamma (IFN-γ), which is secreted by tumor-infiltrating lymphocytes and cancer cells within the tumor, along with other inflammatory mediators [17]. The expression of *PD-L1* at the transcriptional level is a complex process involving various upstream signaling pathways (Figure 1). These pathways play a crucial role in controlling *PD-L1* expression by activating key transcription factors that directly bind to the promoter region of *PD-L1* to promote its expression [18]. The interferon-gamma (IFN-γ)–Janus kinase (JAK) signal transducer and activator of the transcription (STAT) pathway is central to *PD-L1* expression. Upon IFN-γ stimulation, JAK-STAT signaling activates interferon regulatory factor 1 (IRF1), which binds directly to the *PD-L1* promoter, inducing its transcription and suppressing antitumor immunity [19,20]. In addition to IFN-γ signaling, interleukins that utilize the common gamma subunit of the cytokine receptor (IL2RG), specifically IL-2, IL-7, IL-15, and IL-21, activate STAT1 and STAT3 [21]. This activation induces *PD-L1* expression.

The epidermal growth factor receptor (EGFR) signaling pathway can also induce the expression of PD-L1. Upregulation of PD-L1 is mediated by the IL-6/JAK/STAT3 signaling pathway in non-small-cell lung cancer (NSCLC) cells [5]. This suggests a direct link between the activation of EGFR and the regulation of immune responses through PD-L1 expression. In addition, treatment of NSCLC cells with EGFR tyrosine kinase inhibitors (EGFR-TKIs) can downregulate the activation of the IL-6/JAK/STAT3 pathway, which subsequently leads to the reduced expression of PD-L1 [22]. This finding is crucial, as it suggests a potential therapeutic strategy for NSCLC by targeting both the EGFR and PD-L1 pathways.

The binding of the receptor tyrosine kinase c-Met to hepatocyte growth factor (HGF), its ligand, can promote the expression of PD-L1 by activating the Ras-PI3K signaling pathway [23]. Inflammatory cytokines such as IFN-γ, tumor necrosis factor-alpha (TNF-α), and various interleukins (IL-6, IL-10, and IL-27) also play significant roles in PD-L1 expression within the tumor microenvironment (TME) [21]. PD-L1 regulation involves multiple cellular pathways and factors, highlighting the intricate nature of immune regulation.

## 3. Post-Translational Modification (PTM) of PD-L1

PTMs are intricate molecular processes that significantly affect the life cycle of a protein after initial synthesis. These modifications, which occur after protein translation, play a crucial role in dictating the functional journey of a protein and profoundly influence its activity, stability, localization, and interactions [24]. PTMs of PD-L1 play a crucial role in modulating its function, particularly in cancer immunotherapy. These PTMs include glycosylation, phosphorylation, ubiquitination, and acetylation, each contributing to the stability, localization, and functionality of PD-L1. Disruptions in these PTM directly impact immune resistance mediated by PD-L1, making them a focal point in current cancer treatment research.

### 3.1. Glycosylation of PD-L1

Protein glycosylation plays a crucial role in many biological processes, including antibody recognition, cell adhesion, and ligand binding. Dysregulation of glycosylation is associated with various human diseases, including cancer [25].

Li et al. explored the complex interplay between glycosylation, phosphorylation, and ubiquitination and the resulting impact on the stability of PD-L1 [26]. They found that N-linked glycosylation of PD-L1, particularly at N192, N200, and N219, is a critical factor in its stabilization (Figure 2). In contrast, phosphorylation of PD-L1 by glycogen synthase kinase 3β (GSK3β) at T180 and S184 leads to its association with the E3 ubiquitin ligase-transducin repeat-containing protein (β-TrCP), facilitating its degradation [26]. Notably, glycosylation of PD-L1 hinders the phosphorylation by GSK3β, demonstrating a protective effect against PD-L1 degradation. They showed that EGF signaling induces PD-L1 N-linked glycosylation and stabilizes PD-L1, which plays a role in breast cancer cell immunosuppression. This study also demonstrated that the modulation of these pathways, particularly through the inhibition of EGF signaling, offers a promising strategy for enhancing the efficacy of cancer immunotherapy.

The same group also focused on the importance of glycosylation in the PD-L1/PD-1 interaction and its implication in immunosuppression in triple-negative breast cancer (TNBC) [27]. They discovered that glycosylation of PD-L1, mediated by β-1,3-N-acetylglucosaminyltransferase (B3GNT3), is critical for its interaction with PD-1. Downregulation of B3GNT3 or blocking of glycosylated PD-L1 with monoclonal antibodies disrupts the PD-L1/PD-1 interaction, promotes PD-L1 internalization and degradation, and enhances cytotoxic T-cell activity. STT3, a component of the oligosaccharyltransferase complex, is crucial for PD-L1 N-linked glycosylation [28]. N-linked glycosylation is essential for protecting PD-L1 from ubiquitination and proteasome-mediated degradation. Disrupting this glycosylation process, either by STT3 knockdown or pharmacological inhibition, leads to reduced PD-L1 levels in cancer stem cells (CSCs) and enhances their susceptibility to T-cell-mediated killing. In addition, they found that 2-deoxyglucose (2-DG), a glucose analog, inhibited PD-L1 glycosylation. Deglycosylation of PD-L1 results in decreased PD-L1 expression on the cell surface and reduced PD-1 binding, indicating an impairment in the immunosuppressive function of PD-L1. The combination of 2-DG with a poly (ADP-ribose) polymerase (PARP) inhibitor further enhances these effects, demonstrating potent antitumor activity by re-sensitizing cancer cells to T-cell-mediated killing [29].

These studies underscore the significance of glycosylation in the regulation of PD-L1 and its impact on immune evasion mechanisms in cancer. By disrupting these glycosylation pathways, it may be possible to enhance the effectiveness of immune checkpoint inhibitors and develop more effective treatments for various cancers.

### 3.2. Phosphorylation of PD-L1

Protein phosphorylation, a common post-translational modification, occurs when a protein kinase covalently adds a phosphate group to an amino acid. This modification is crucial in various human diseases, including cancer.

PD-L1 is phosphorylated by several kinases that significantly affect its function and stability. Phosphorylation of PD-L1 by GSK3β is differentially regulated in glycosylated and non-glycosylated forms of PD-L1. In the non-glycosylated form, GSK3β-mediated phosphorylation results in PD-L1 binding to β-TrCP, facilitating its degradation. Conversely, glycosylation of PD-L1 impedes GSK3β binding, thus inhibiting its phosphorylation and subsequent degradation [26].

AMP-activated protein kinase (AMPK) phosphorylates PD-L1 at Serine 195 (S195), causing abnormal glycosylation and accumulation in the endoplasmic reticulum (ER) [30]. Aberrant glycosylation blocks PD-L1’s transition to the Golgi apparatus, leading to its degradation via the ER-associated degradation (ERAD) pathway. Furthermore, IL-6-activated JAK1 specifically phosphorylates PD-L1 at Tyr112, which is a critical step in the functional regulation of the protein [31]. This phosphorylation facilitates the recruitment of ER-associated N-glycosyltransferase STT3A, which is essential for PD-L1 glycosylation and subsequent stability.

Zhang et al. highlighted the role of NEK2, or never in mitosis gene A-related kinase 2, in the phosphorylation of PD-L1, specifically at the T194/T210 residues. This phosphorylation contributes to the maintenance of PD-L1 stability by preventing its degradation via the ubiquitin–proteasome pathway in the ER lumen [32]. NEK2 deficiency leads to decreased PD-L1 expression and increased lymphocyte infiltration, thereby enhancing the anticancer immune response, particularly in PD-L1-targeted pancreatic cancer immunotherapy. Furthermore, casein kinase 2 has been observed to phosphorylate PD-L1 at Thr285 and Thr290, leading to the stabilization of PD-L1 in tumors and dendritic cells [33].

These findings emphasize the importance of differential regulation of PD-L1 phosphorylation, with particular emphasis on its glycosylated and non-glycosylated forms. This differential regulation plays a pivotal role in determining the efficacy of cancer immunotherapies, highlighting the potential of targeted therapeutic strategies to manipulate PD-L1 phosphorylation.

### 3.3. Acetylation of PD-L1

Protein acetylation is an important PTM in eukaryotic organisms. This process involves the transfer of an acetyl group from acetyl coenzyme A (acetyl-CoA) to a designated location on the polypeptide chain [34]. The acetyl group provided by the metabolite acetyl-Co A may be attached either co-translationally or post-translationally to the α-amino group at the N-terminus of proteins or to the ε-amino group on lysine residues. Different types of acetyltransferases, specifically N-terminal and lysine acetyltransferases, catalyze these processes.

Horita et al. first demonstrated that treatment with EGF leads to the acetylation of PD-L1 in conjunction with its phosphorylation and ubiquitination [35]. PD-L1 undergoes acetylation at Lys263 within its cytoplasmic domain, a modification catalyzed by p300 acetyltransferase [36]. Acetylation plays a crucial role in its translocation from the cell membrane to the nucleus, in contrast to glycosylation. While glycosylation inhibits PD-L1’s ubiquitination-mediated degradation, enhancing its stability and interaction with PD-1 for tumor immune escape, acetylation facilitates its nucleoplasmic translocation. In the nucleus, acetylated PD-L1 binds to DNA and influences the expression of immune response genes, thereby modulating the antitumor immune response [36]. Notably, the nuclear translocation of PD-L1 is independent of the glycosylation status. The dynamic regulation of PD-L1 acetylation and deacetylation, also involving histone deacetylase 2 (HDAC2), highlights potential targets for enhancing antitumor efficacy through the pharmacological modulation of PD-L1 acetylation, thereby impeding its nuclear translocation and boosting the effectiveness of PD-1 blockade therapies.

### 3.4. Palmitoylation of PD-L1

Protein palmitoylation, primarily known as S-palmitoylation, involves the reversible covalent attachment of palmitic acid to cysteine residues and, less frequently, to serine and threonine residues (O-palmitoylation) in proteins through a thioester bond. This modification, usually facilitated by DHHC protein acyltransferases (DHHC-PATs) with an Asp-His-His-Cys (DHHC) active center, significantly influences protein membrane anchoring, transport, degradation, and subcellular trafficking [37]. Furthermore, palmitoylation plays a crucial role in modulating protein–protein interactions and the hydrophobicity of the palmitoyl group aids in the membrane attachment of proteins.

Yang et al. first reported that PD-L1 undergoes palmitoylation at Cys272 in its cytosolic domain, which is crucial for its stability [38]. The study also identified zinc finger DHHC-type palmitoyltransferase 9 (ZDHHC9), a member of the DHHC protein acyltransferase family, as a key enzyme involved in PD-L1 palmitoylation. Inhibition of palmitoylation makes tumor cells more susceptible to immune attacks, suggesting potential therapeutic strategies for targeting palmitoylation in breast cancer. Yao et al. also identified ZDHHC3 as the main acetyltransferase required for the palmitoylation of PD-L1 and showed that the inhibition of PD-L1 palmitoylation via 2-bromopalmitate or the silencing of DHHC3-activated antitumor immunity in vitro and in mice bearing MC38 tumor cells [39]. They designed a competitive inhibitor of PD-L1 palmitoylation that decreased PD-L1 expression in tumor cells to enhance T-cell immunity against tumors. These findings suggest novel strategies for overcoming PD-L1-mediated immune evasion in cancer. Another study demonstrated that PD-L1 undergoes significant palmitoylation in cisplatin-resistant bladder cancer cells [40]. This modification of PD-L1 contributes to its stability and may influence the effectiveness of cisplatin treatment. The pharmacological inhibition of fatty acid synthase (FASN), which is involved in palmitoylation, leads to reduced PD-L1 palmitoylation and expression. This suggests a potential therapeutic strategy for targeting the FASN-PD-L1 axis in bladder cancer. These findings suggest that targeting PD-L1 palmitoylation could be a promising strategy in cancer therapy, potentially enhancing the efficacy of existing treatments such as immune checkpoint inhibitors.

### 3.5. SUMOylation of PD-L1

Small ubiquitin-like modifier (SUMO) proteins are similar to ubiquitin in their folded structure but possess approximately 20% homology with the amino acid sequence of ubiquitin [41]. SUMOylation, the process in which SUMO proteins are covalently bonded to certain lysine residues on a target protein, is crucial for the regulation of a variety of cellular functions [42]. This regulation is achieved through its influence on the functional characteristics of the proteins.

Ma et al. found that tripartite motif-containing 28 (TRIM28) binds to and stabilizes PD-L1 by promoting PD-L1 SUMOylation [43]. TRIM28 is composed of a TRIM structure at the N-terminus, a plant homeodomain (PHD), and a bromodomain at the C-terminus. It is a classical RING-type E3 ubiquitin ligase [44]. This study identified TRIM28 as a significant regulator of PD-L1 stability, using genome-wide CRISPR-Cas9-based screening. It stabilizes PD-L1 in gastric cancer by inhibiting its ubiquitination and promoting SUMOylation. Additionally, PD-L1 can be post-translationally modified by SUMO2, and the ectopic expression of TRIM28 enhances this SUMO2 modification of PD-L1 [43].

## 4. Ubiquitination and Deubiquitination of PD-L1

Ubiquitination and deubiquitination represent critical PTMs of PD-L1 that profoundly influence the immune response, cancer progression, and therapeutic strategies. Understanding this process provides valuable insights into the mechanisms of immune evasion by tumors and offers potential strategies for therapeutic interventions.

### 4.1. General Principles of the Ubiquitin–Proteasome Pathway

The ubiquitin–proteasome pathway (UPP) is a complex and fundamental cellular mechanism responsible for protein degradation and regulation in eukaryotic cells. This system plays a critical role in maintaining cellular homeostasis and governs various biological functions [45,46]. At its core is the dynamic and reversible process of ubiquitination complemented by DUBs, which provide an additional layer of regulation and specificity.

UPP begins with the activation of ubiquitin, a small regulatory protein, by the E1 enzyme (ubiquitin-activating enzyme) (Figure 3). This ATP-dependent step marks the initiation of ubiquitination and sets the stage for the subsequent processes. The activated ubiquitin is then transferred to an E2 enzyme (a ubiquitin-conjugating enzyme), which primes it for the crucial action of E3 enzymes (ubiquitin ligases). E3 ligases play a pivotal role in UPP specificity by recognizing target proteins and facilitating ubiquitin transfer from E2 to these targets [47]. Selective tagging of proteins with ubiquitin, particularly through polyubiquitination, marks them for degradation.

Central to the regulatory mechanisms of UPP are the DUBs, a diverse group of enzymes that remove ubiquitin from substrate proteins. DUBs have several critical functions: they can rescue proteins from degradation by removing ubiquitin tags, processing ubiquitin precursors, and recycling ubiquitin by disassembling polyubiquitin chains [48]. Their action is essential for the balance and fidelity of protein ubiquitination, influencing the fate of proteins and the outcome of ubiquitin signaling.

Once tagged for degradation, proteins are recognized by the 26S proteasome, which is a complex protein-degrading machinery composed of 20S core and 19S regulatory particles [49]. The 19S subunit binds to the ubiquitinated proteins, unfolds them, and guides them into the 20S core, where they are enzymatically broken down into peptides. This degradation process is crucial for regulating protein levels and recycling amino acids within the cells.

The significance of the ubiquitin–proteasome system extends to disease processes, with dysregulation of either the ubiquitination process or DUB activity contributing to the accumulation of abnormal proteins and various disorders, including cancer and immune system dysfunction [50]. Consequently, proteasomes and DUBs have emerged as potential therapeutic targets [51,52]. Proteasome inhibitors such as bortezomib are used in treating specific cancers, while modulating DUB activity presents new avenues for drug discovery and therapeutic interventions.

### 4.2. E3 Ubiquitin Ligases in PD-L1 Regulation

In this section, we will explore the latest research regarding the molecular characteristics and functions of specific E3 ubiquitin ligases that are involved in the ubiquitination of PD-L1 (Table 1).

#### 4.2.1. β-TrCP

β-TrCP, an F-box protein, serves as the core component of the E3 ubiquitin ligase within the SCF complex (comprising SKP1, CUL1, and F-box proteins). This complex orchestrates the ubiquitination of various substrates via the ubiquitin–proteasome pathway [53].

As elucidated earlier, the phosphorylation of PD-L1 by GSK3β is a pivotal event that facilitates its interaction with β-TrCP. This interaction culminates in the ubiquitination of PD-L1, which ultimately leads to its degradation by the 26S proteasome [26]. This molecular interplay takes place at the specific β-TrCP-binding motif on PD-L1. However, PD-L1’s glycosylation at sites N192, N200, and N219 creates a spatial barrier that restricts GSK3β from accessing and phosphorylating PD-L1, effectively stabilizing PD-L1.

Notably, in breast cancer cells, the activation of EGF inactivates GSK3β, thereby promoting the stabilization of PD-L1. Conversely, agents, such as gefitinib, counteract this effect by suppressing the EGF pathway. This intervention enhances antitumor T-cell immunity and augments the efficacy of anti-PD-1 therapy [26].

The intricate interplay between β-TrCP, GSK3β, and PD-L1 presents therapeutic opportunities. Inhibiting β-TrCP or deactivating GSK3β can impede PD-L1 ubiquitination, consequently increasing its stability. While this stabilization may contribute to cancer immunosuppression, it also opens up avenues for strategies aimed at improving the efficacy of immunotherapies.

#### 4.2.2. STUB1

CMTM6 is a type 3 transmembrane protein belonging to the CKLF-like MARVEL transmembrane domain-containing (CMTM) family. CMTM6 was identified as a critical regulator of PD-L1 via a whole-genome CRISPR/Cas9 deletion library screen [54]. CMTM6 binds to PD-L1 and maintains its cell surface expression. The interaction between CMTM6 and PD-L1 occurs at the plasma membrane and in recycling endosomes, where CMTM6 prevents PD-L1 from being targeted for lysosomal degradation. In addition, the depletion of CMTM6 leads to a significant decrease in PD-L1 levels, thereby alleviating the suppression of tumor-specific T-cell activity.

Further investigations revealed that stress-inducible protein-1 (STIP1) homology and U-Box-containing protein 1 (STUB1) play a critical role in the regulation of PD-L1 expression. In the absence of CMTM6, an increase in the ubiquitination of PD-L1 was observed, implying that CMTM6 acts as a protective shield, shielding PD-L1 from STUB1-mediated ubiquitination. Notably, when STUB1 is deleted, PD-L1 levels increase, particularly in cells deficient in CMTM6, underscoring STUB1’s active involvement in targeting PD-L1 for degradation [55].

Recently, pyridoxal, a member of the vitamin B6 family, emerged as a potent suppressor of PD-L1 expression [56]. This discovery surfaced through a metabolite library screen designed to identify natural metabolites capable of modulating PD-L1 expression. This study revealed that the absence of STUB1 effectively counteracted the pyridoxal-induced reduction in endogenous PD-L1 levels. This finding underscores the importance of STUB1 in pyridoxal-mediated PD-L1 degradation. Functionally, pyridoxal demonstrated the ability to enhance T-cell killing activity by disrupting the PD-L1/PD-1 signaling pathway. Although the precise molecular mechanisms governing pyridoxal-mediated PD-L1 degradation require comprehensive elucidation, pyridoxal is a promising candidate for synergistic use in cancer immunotherapies.

#### 4.2.3. SPOP

The speckle-type POZ protein (SPOP) is a component of the Cullin-RING E3 ubiquitin ligase complex and has been shown to regulate PD-L1 expression. Zhang et al. demonstrated that cyclin D–cyclin-dependent kinase 4 (CDK4) and the Cullin 3 SPOP E3 ligase complex play pivotal roles in controlling PD-L1 abundance through proteasome-mediated degradation [57]. Specifically, cyclin D-CDK4 phosphorylates and stabilizes SPOP, which in turn leads to polyubiquitination and downregulation of the PD-L1 protein, particularly in the late G1 and S phases of the cell cycle. Furthermore, treatment with the CDK4/6 inhibitor palbociclib in syngeneic colon cancer mouse models not only increased PD-L1 protein expression but also synergized with anti-PD-1 therapy, enhancing therapeutic efficacy. This suggests that targeting the regulatory mechanisms of PD-L1 polyubiquitination across different cellular compartments or cell cycle phases could significantly improve the efficacy of immunotherapy in patients with cancer.

Additionally, Meng et al. found that the phosphorylation of moesin (MSN) by Rho-associated protein kinase (ROCK) stabilized PD-L1 protein levels [58]. Phosphorylated MSN competes with the E3 ubiquitin ligase SPOP to bind PD-L1. Inhibition of ROCK using a Y-27632 inhibitor or silencing MSN decreased PD-L1 expression, resulting in T-cell activation both in vitro and in vivo. This suggests that the ROCK-MSN pathway plays a vital role in the regulation of PD-L1 and has implications in breast cancer immunotherapy.

#### 4.2.4. HRD1

Hydroxymethylglutaryl-coenzyme A (HMG-CoA) reductase-degrading protein 1 (HRD1) has been identified as a key ERAD factor that directly catalyzes ubiquitin conjugation onto unfolded or misfolded proteins for proteasomal degradation [59]. Cha et al. explored the role of HRD1 and the effects of metformin on the stability of PD-L1 [30]. Metformin directly phosphorylates serine 195 (S195) of PD-L1 by activating AMPK. This phosphorylation leads to the abnormal glycosylation of PD-L1, resulting in its accumulation in the ER via the ERAD pathway. Further examination showed that the phosphorylation of PD-L1 at S195 altered its glycan structure, blocking its translocation from the ER to the Golgi. HRD1 knockdown reduced metformin-induced PD-L1 ubiquitination, further corroborating HRD1’s role in the degradation of PD-L1 degradation via the ERAD pathway.

#### 4.2.5. RNF144A

Ring finger 144A (RNF144A) is a member of the RNF144 family of E3 ubiquitin ligases that have a RING (RBR) domain at the N-terminus [60]. This study revealed that RNF144A deficiency in mice leads to increased levels of PD-L1, suggesting that RNF144A mediates the ubiquitination and degradation of PD-L1 [61]. This study indicated that the absence of RNF144A stabilizes PD-L1, contributing to the reduction of tumor-infiltrating CD8^+^ T-cell populations in bladder tumors. The interaction between RNF144A and PD-L1 highlights the therapeutic significance of targeting RNF144A in cancer treatment, particularly in bladder cancer, where PD-L1 plays a crucial role in immune evasion.

#### 4.2.6. NEDD4

Neuronal precursor cell-expressed developmentally downregulated 4 (NEDD4), a prominent member of the HECT family of E3 ubiquitin ligases [62], plays a central role in the intricate regulation of PD-L1. Jing et al. unveiled a captivating revelation: the interplay between fibroblast growth factor receptor 3 (FGFR3) and PD-L1 via NEDD4 [63]. When FGFR3 is activated, often due to mutations or overexpression in bladder cancer, a cascade of events is triggered. FGFR3 phosphorylates NEDD4, thereby establishing the stage for a significant chain reaction. Phosphorylated NEDD4, in turn, catalyzes the ubiquitination of PD-L1, marking it for subsequent degradation. This process has profound implications for immune surveillance in bladder cancer, given PD-L1’s well-documented role in dampening the antitumor activity of CD8^+^ T cells.

From a therapeutic perspective, targeting FGFR3, and consequently, NEDD4 and PD-L1, has emerged as a promising strategy to amplify the efficacy of immune checkpoint therapies such as anti-PD-1, particularly in cases involving FGFR3 activation. This study underscores the crucial link between targeted therapy and immune surveillance, with NEDD4 as a pivotal regulator of this intricate pathway.

#### 4.2.7. FBXO22

F-box-only protein 22 (FBXO22) is a member of the F-box family and a subunit of the ubiquitin E3 ligase complex SCF (SKP1–Cullin–F-box) [64]. De et al. found that high expression of FBXO22 leads to decreased PD-L1 levels and sensitizes cancer cells to DNA-damaging therapies, such as ionizing radiation and cisplatin [65]. This effect was partly caused by the regulation of PD-L1 levels by CDK5 because CDK5 inhibition increased FBXO22 levels, leading to reduced PD-L1 expression and increased sensitivity to DNA damage. This study highlights the therapeutic potential of targeting the CDK5-FBXO22-PD-L1 pathway in NSCLC treatment, suggesting that CDK5 inhibitors could enhance the efficacy of immunotherapy and DNA-damaging therapies.

#### 4.2.8. TRIM21

TRIM21 is a member of the tripartite motif-containing protein (TRIM) family of RING-type E3 ubiquitin ligases, but uniquely among the TRIM family, TRIM21 has high-affinity antibody-binding activity [66]. Zhang et al. found that TRIM21 directly binds to PD-L1, and its expression is decreased in radiation-resistant NSCLC cells. Dihydroartemisinin treatment enhances radiation sensitivity and displays an inverse correlation by reducing PD-L1 levels and elevating TRIM21 expression, suggesting a potential pathway for overcoming radiation resistance in lung cancer cells by targeting the PD-L1/TRIM21 axis [67].

Gao et al. explored the role of CDK5 in lung adenocarcinoma and found that CDK5 inhibition led to a reduction in PD-L1 protein levels via the ubiquitination–proteasome pathway mediated by TRIM21 [68]. This reduction in PD-L1 expression enhances antitumor immunity, as demonstrated in lung cancer cell lines and mouse models. This study highlights CDK5 as a potential therapeutic target, in combination with immunotherapy, for advanced lung adenocarcinoma.

Long non-coding RNA LINC02418 downregulates PD-L1 expression and enhances T-cell-induced apoptosis in a TRIM21-dependent manner in NSCLC [69]. This study established the physiological and pathological significance of LINC02418 and TRIM21 in regulating the PD-L1-mediated immune checkpoint therapeutic response. This study provides a new perspective on the PD-L1/TRIM21 axis and offers a potential pharmaceutical intervention target for patients with NSCLC undergoing anti-PD-L1 treatment.

**Table 1 ijms-25-02939-t001:** Role of E3 ubiquitin ligases in the regulation of PD-L1 and its effects on cancer.

E3 Ligase	Mechanism	Types of Cancer	Effects	References
β-TrCP	binds to PD-L1 after GSK3β phosphorylation, leading to PD-L1’s ubiquitination and degradation	Breast cancer	promotes antitumor T-cell immunity and enhances the effectiveness of anti-PD-1 therapy	[28]
STUB1	induces ubiquitination and destabilization of PD-L1	Lung cancer	negatively modulates immune activity	[57]
SPOP	stabilized by CDK4 phosphorylation, downregulates PD-L1 protein, especially during the late G1 and S phases of the cell cycle	Colon cancer Breast cancer	increases T-cell activation	[59,60]
HRD1	HRD1 knockdown diminishes metformin-induced PD-L1 ubiquitination	Breast cancer	enhances antitumor CTL immunity	[32]
RNF144A	interacts with PD-L1 and promotes poly-ubiquitination and degradation of PD-L1	Bladder cancer	Enhances tumor-infiltrating CD8^+^ T-cell populations	[63]
NEDD4	NEDD4, phosphorylated upon FGFR3 activation, catalyzes the ubiquitination and degradation of PD-L1	Bladder cancer	Increases the antitumor activity of CD8^+^ T cells	[65]
FBXO22	FBXO22’s high expression reduces PD-L1 levels, enhancing cancer cell sensitivity to DNA-damaging therapies	NSCLC	enhances the efficacy of immunotherapy	[67]
TRIM21	enhances ubiquitination-mediated proteasomal degradation of PD-L1	NSCLC	enhances T-cell-induced apoptosis	[70,71]

β-TrCP, β-transducin repeat-containing protein; PD-L1, programmed death ligand 1; GSK3β, glycogen synthase kinase 3β; STUB1, stress-inducible protein-1 (STIP1) homology and U-box containing protein 1; SPOP, speckle-type POZ protein; CDK, cyclin-dependent kinase; HRD1, hydroxymethylglutaryl-coenzyme A (HMG-CoA) reductase-degrading protein 1; CTL, cytotoxic T lymphocyte; RNF144A, ring finger 144A; CD8, cluster of differentiation 8; NEDD4, neuronal precursor cell-expressed developmentally downregulated 4; FBXO22, F-box-only protein 22; NSCLC, non-small-cell lung cancer; TRIM21, tripartite motif-containing protein 21.

### 4.3. Deubiquitinating Enzymes in PD-L1 Regulation

DUBs play a pivotal role in PD-L1 regulation (Table 2). DUBs are classified based on their catalytic mechanisms and structural features and encompass various superfamilies, including ubiquitin-specific protease (USP), ubiquitin C-terminal hydrolase (UCH), ovarian tumor protease (OTU), Machado–Josephin domain superfamily (MJD), and JAMMs (JAB1/MPN/Mov34) metalloproteases. These enzymes specifically cleave ubiquitin from ubiquitinated substrates to regulate crucial cellular processes [70,71,72,73,74].

#### 4.3.1. OTUB1

OUT domain-containing ubiquitin aldehyde-binding protein 1 (OTUB1), a member of the OTU subfamily of DUBs, is a proteasome-associated DUB pivotal for the regulation of PD-L1 [75]. OTUB1’s remarkable function lies in its ability to remove K48-linked ubiquitin chains from PD-L1, thereby inhibiting PD-L1 degradation via the ERAD pathway [76].

The depletion of OTUB1 has compelling consequences, leading to a reduction in PD-L1 levels. This, in turn, enhances CD8^+^ T-cell infiltration and augments antitumor immunity. This intricate relationship is further exemplified in NSCLC, in which the circular RNA insulin-like growth factor 2 mRNA-binding protein 3 (circIGF2BP3) modulates OTUB1 levels, counteracts the ubiquitinated degradation of PD-L1, and suppresses CD8^+^ T-cell function, ultimately facilitating immune evasion [77]. These findings open new avenues for potential therapeutic targets, highlighting the role of modulating OTUB1 or circIGF2BP3 levels in enhancing the efficacy of cancer immunotherapies.

#### 4.3.2. CSN5

The constitutive photomorphogenesis 9 (COP9) signalosome 5 (CSN5), also known as c-Jun activation domain-binding protein-1 (Jab1), contains a JAMM metalloprotease motif located within the JAMM subfamily [78]. CSN5 was the first identified DUB responsible for the deubiquitination and stabilization of PD-L1 [79].

CSN5 induction, orchestrated by NF-κB p65, proves indispensable for TNF-α-mediated stabilization of PD-L1 in cancer cells. The insights gained from this research highlight the potential of curcumin in cancer treatment, particularly as a complementary approach to existing immunotherapies, by targeting CSN5-mediated stabilization of PD-L1. Additionally, natural compounds, such as shikonin, have been shown to inhibit the activity and expression of CSN5, thereby influencing PD-L1 stability [80]. These findings suggest that shikonin compounds could serve as valuable additives to anticancer treatments, offering a means to modulate the PD-L1 pathway and bolster the immune response against pancreatic cancer.

#### 4.3.3. USP2

Ubiquitin-specific protease 2 (USP2), a DUB cysteine protease and a well-established member of the USP subfamily, boasts a repertoire of biological effects ranging from cell cycle progression to carcinogenesis and circadian rhythm regulation [81]. Recent breakthroughs have identified USP2 as a novel regulator of PD-L1 stabilization, which was validated using single-guide RNAs (sgRNAs) screening systems [82].

USP2’s interaction with PD-L1 is a testament to its significance. The ubiquitin-binding domain of USP2 directly engages with the intracellular domain (ICD) of PD-L1. Functionally, USP2 deubiquitinates K48-linked polyubiquitination of PD-L1, resulting in increased PD-L1 abundance in colorectal and prostate cancer cells. Depletion of USP2 results in ERAD-dependent degradation of PD-L1, leading to reduced PD-L1/PD-1 interactions and enhanced susceptibility of cancer cells to T-cell-mediated killing. This phenomenon enhances antitumor immunity, which is characterized by increased CD8^+^ T-cell infiltration and decreased immunosuppressive infiltration of myeloid-derived suppressor cells (MDSCs) and Tregs.

High USP2 expression in colorectal cancer is associated with reduced antitumor immunity and poor clinical outcomes, highlighting the USP2’s negative regulatory role in T-cell immunity [82]. This study suggests that USP2 inhibitors can be used in combination with existing PD-L1-based therapies to increase their efficacy.

#### 4.3.4. USP5

USP5 has emerged as a prominent PD-L1 deubiquitinase in NSCLC cells. It interacts directly with PD-L1 and stabilizes its expression [83]. Elevated USP5 levels in NSCLC tissues are correlated with increased PD-L1 levels and poor patient prognosis. Notably, USP5 knockdown in a Lewis lung carcinoma mouse model resulted in reduced tumor growth, emphasizing USP5’s role in PD-L1 regulation and the promotion of lung cancer progression.

Furthermore, the connection between PD-L1 expression and epithelial–mesenchymal transition (EMT) has become increasingly evident [84,85]. Cai et al. revealed that USP5 directly interacts with Twist1 and enhances its protein stability, facilitating the EMT process [86]. This interaction underscores the significance of the USP5-Twist1 axis in cancer biology, particularly in the context of tumor metastasis and progression.

#### 4.3.5. USP7

USP7, also known as herpesvirus-associated ubiquitin-specific protease (HAUSP), is a member of the USP subfamily of deubiquitinases that plays a crucial role in various tumor types [87,88]. Its involvement in numerous substrates linked to cancer progression renders it a potential therapeutic target [89]. Recent studies have highlighted the positive association between USP7 expression and PD-L1 protein levels in gliomas and gastric cancer [90,91].

In gliomas, USP7 is consistently overexpressed and intricately linked to PD-L1 expression. USP7 stabilizes PD-L1 by deubiquitination and contributes to the immune evasion of glioma cells. The suppression of USP7 in glioma cells hampers their growth and induces apoptosis. It also boosts CD8^+^ T-cell proliferation, thereby reducing the ability of these cells to evade the immune system. However, this effect is reversed by PD-L1 overexpression, highlighting the dynamic interplay between USP7 and PD-L1 in glioma immune escape mechanisms [90].

Similarly, USP7 directly interacts with PD-L1 in GC, enhancing its stability and affecting cancer cell proliferation. Silencing USP7 leads to decreased PD-L1 expression and increased T-cell-mediated cancer cell destruction [91]. These findings indicate the potential of USP7 inhibition in cancer treatment, particularly in strategies targeting the PD-L1/PD-1 pathway.

Dai et al. demonstrated that targeting USP7 with siRNA or inhibitors altered M2 macrophages and promoted CD8^+^ T-cell proliferation [92]. In mouse models of Lewis lung carcinoma, USP7 inhibitors slowed tumor growth, increased M1 macrophages, and enhanced IFN-γ^+^ CD8^+^ T-cell tumor infiltration. Furthermore, USP7 inhibitors elevated PD-L1 expression in tumors and, when combined with PD-1 blockade, enhanced antitumor responses. A selective USP7 inhibitor, P5091, effectively inhibits CT26 xenograft growth in mice, similar to anti-PD-1 antibody treatment [93].

#### 4.3.6. USP8

In pancreatic cancer, USP8 is notably upregulated and is positively correlated with disease progression [94]. Elevated USP8 expression was associated with increased PD-L1 expression. Targeting USP8 enhances the efficacy of anti-PD-L1 therapy by reducing PD-L1 protein degradation and subsequently improving the activation of cytotoxic T lymphocytes and overall antitumor immunity. A combined approach using USP8 and PD-L1 inhibitors has the potential to reduce tumor growth and increase the effectiveness of CD8^+^ T-cell-mediated cancer cell destruction.

Xiong et al. underscored the significant effect of DUBs-IN-2, a USP8 inhibitor, on PD-L1 protein levels in various cancer cell lines [95]. Notably, the inhibition of USP8 in lung squamous cancer tissues leads to an increase in PD-L1 protein levels, indicating a negative correlation between USP8 and PD-L1 in these cancer tissues. Mechanistically, USP8 influenced PD-L1 degradation by altering its ubiquitination, specifically by removing K63-linked ubiquitin chains and promoting K48-linked ubiquitination. USP8 also modulates immune responses by affecting the NF-κB signaling pathway through its interaction with tumor necrosis factor receptor-associated factor 6 (TRAF6). The combined use of a USP8 inhibitor and anti-PD-L1/PD-1 treatment significantly reduced tumor growth and improved survival in mouse colon cancer models. This study highlights the potential of USP8 inhibitors as a therapeutic strategy for enhancing the effectiveness of immunotherapy.

#### 4.3.7. USP9X

The X-linked deubiquitinase USP9X has been implicated in various cellular processes, including cell growth, division, and death, which affect cancer development. Its role varies between tumor promotion and suppression, depending on the cancer type [96].

In oral squamous cell carcinoma (OSCC), USP9X interacts with PD-L1, leading to its deubiquitination and stabilization of PD-L1. Suppression of USP9X in OSCC has emerged as a potential strategy to inhibit tumor growth, suggesting that USP9X is a potential therapeutic target for this cancer type [97].

#### 4.3.8. USP18

USP18, a unique member of the USP subfamily, is exclusive for the deconjugation of the ubiquitin-like protein interferon-stimulated gene 15 (ISG15) from target proteins with no activity toward ubiquitin [98]. USP18 is an IFN-stimulated gene product and a negative regulator of type I IFN signaling [99]. Recent studies have highlighted its pivotal role in bladder cancer, particularly in the regulation of PD-L1 expression.

USP18 is upregulated in tumor tissues and contributes to the malignant phenotype of bladder cancer cells [100,101]. A genetic variant, rs62483508 G>A, located in the microRNA response elements (MREs) of the long non-coding RNA (lncRNA) bladder cancer cell cytoplasm-enriched abundant transcript 4 (BCCE4), has been identified as a key factor in bladder cancer susceptibility [102]. Mechanistically, the lncRNA BCCE4 A allele loses a binding site for miR-328-3p, resulting in decreased USP18 expression, owing to its inability to function as an miRNA sponge. This downregulates PD-L1 levels and restores the antitumor immune response in bladder cancer. A positive correlation was established between USP18 levels and PD-L1 expression in bladder cancer tissues. This correlation underscores the regulatory role of USP18 in the modulation of PD-L1 expression. In bladder cancer, USP18 augments the removal of ISG15 from PD-L1, enhancing the stability of PD-L1. The increased stability of PD-L1 is a key factor in the immune evasion of bladder cancer cells. Furthermore, the enrichment of the lncRNA BCCE4 in exosomes from bladder cancer plasma, tissues, and cells highlights its potential as a biomarker for disease progression. These insights not only enhance our understanding of the genetic factors in bladder cancer but also open new avenues for the development of targeted therapies and diagnostic tools, particularly in the context of the PD-L1/PD-1 axis and its regulation by USP18.

#### 4.3.9. USP20

USP20, also known as von Hippel–Lindau (pVHL)-interacting deubiquitinating enzyme 2 (VDU2), was initially identified as a deubiquitinase associated with von Hippel–Lindau (VHL) syndrome [103,104]. This enzyme causes deubiquitination of various substrates, including type II deiodinase (D2), hypoxia-inducible factor-1α (HIF-1α), β2-adrenergic receptor (β2-AR), TRAF6, and pyruvate kinase M2 (PKM2). Consequently, it plays a significant role in thyroid hormone activation, cell cycle regulation, inflammatory responses, tumorigenesis, tumor progression, and other vital biological processes.

Wang et al. demonstrated that USP20 directly interacts with PD-L1 [105]. This interaction is modulated by the long non-coding RNA tissue differentiation-inducing non-protein coding RNA (TINCR). TINCR recruits DNA (cytosine-5)-methyltransferase 1 DNMT1, leading to the methylation of miR-199a-5p, thereby reducing its transcription. In addition, TINCR acts as a molecular sponge for miR-199a-5p in the cytoplasm, thereby stabilizing USP20 mRNA. This, in turn, promotes PD-L1 expression by reducing its ubiquitination in breast cancer, ultimately leading to the stabilization of PD-L1.

#### 4.3.10. USP21

USP21 is a nuclear/cytoplasmic shuttle protein with a catalytic deubiquitinase (DUB) domain at its C-terminus. It has been implicated in promoting cancer, contributing to tumor growth, enhancing the stem-like properties of cancer cells, and facilitating metastasis in various types of tumors [106,107].

Yang et al. demonstrated a direct interaction between USP21 and PD-L1 [108]. USP21 interacts with PD-L1 by removing its polyubiquitin chains, resulting in the stabilization of PD-L1. Moreover, cancer-related mutations in PD-L1, particularly at Asp276, enhance the USP21-mediated deubiquitination of PD-L1. In lung cancer, particularly lung squamous cell carcinoma, there is often a simultaneous increase in USP21 amplification and PD-L1 expression. USP21 also prevents the generation of Th1-like Treg cells, which play a key role in tumor immune evasion [109].

Deng et al. explored the effects of gallic acid on USP21 and its implications for immunotherapy [110]. Gallic acid suppressed USP21 transcription, leading to the degradation of FOXP3 and PD-L1 proteins. Weakening of the suppressive function of regulatory T (Treg) cells enhances the effectiveness of immune checkpoint blockade therapy in colorectal cancer by promoting the formation of T-helper-1-like Treg cells. These findings suggest that by targeting USP21, gallic acid has the potential to potentiate the efficacy of immunotherapies for cancer.

#### 4.3.11. USP22

USP22 is a DUB and subunit of the Spt-Ada-Gcn5-acetyltransferase (SAGA) transcriptional coactivator complex that functions by removing ubiquitin from target proteins, thereby regulating the transcription of downstream genes [111]. Studies across various cancer types have consistently shown that high USP22 expression is associated with enhanced cancer cell survival, immune evasion, and poor prognosis. USP22 is not only indicative of aggressive cancer types but also actively contributes to cancer progression [112,113].

USP22 has also been identified as a PD-L1 deubiquitinase [114,115]. USP22 directly interacts with the C-terminus of PD-L1, thereby inducing its deubiquitination and stabilization. Its potential influence on the immune system has been studied in liver cancer [114]. In NSCLC, USP22 stabilizes CSN5 through deubiquitination and enhances PD-L1-CSN5 interaction, further influencing PD-L1 functionality [115]. The depletion of USP22 in cancer models leads to reduced tumor growth and increased T-cell cytotoxicity. A significant positive correlation between USP22 and PD-L1 expression was also observed in human lung cancer tissues.

Recently, Huang et al. demonstrated that Usp22 deficiency in bone marrow-derived cells leads to the reduced production of type I and II interferons, which do not significantly affect viral replication [114]. However, this deficiency results in limited PD-L1 expression, causing an increase in functional, virus-specific CD8^+^ T cells and subsequent rapid death in Usp22-deficient mice. Depletion experiments on these CD8^+^ T cells indicated their critical role in the enhanced lethality observed in Usp22-deficient mice. Overall, this study concluded that the absence of Usp22 triggered a pathological CD8^+^ T-cell response, leading to severe disease in mice.

These findings highlight the critical role of USP22 in cancer progression and immune system regulation. Evidence suggests that targeting USP22 could improve the efficacy of immune checkpoint blockade therapies, especially in cancers in which PD-L1-mediated immune evasion is a key factor.

**Table 2 ijms-25-02939-t002:** Role of deubiquitinating enzymes (DUBs) in the regulation of PD-L1 and their effects on cancer.

DUBs	Mechanism	Types of Cancer	Effects	References
OTUB1	removes K48-linked ubiquitin on PD-L1, thus inhibiting its degradation	lung cancer	decreases CD8^+^ T-cell infiltration and antitumor immunity	[78,79]
CSN5	removes K48-linked ubiquitin on PD-L1	breast cancer	promotes immune evasion	[81,82]
USP2	removes K48-linked ubiquitin on PD-L1, leading to increased PD-L1 abundance	colon cancer, prostate cancer	confers resistance to cancer cells against T-cell-mediated killing	[84]
USP5	stabilizes PD-L1 through interaction and deubiquitination	NSCLC	inhibits CD8^+^ T-cell cytotoxicity	[85]
USP7	stabilizes PD-L1 through interaction and deubiquitination	glioma gastric cancer	inhibits apoptosis and boosts CD8^+^ T-cell proliferation suppresses M1 macrophages and IFN-γ^+^ CD8^+^ T-cell tumor infiltration	[92,93]
USP8	removes K63-linked ubiquitin on PD-L1 stabilizes PD-L1 through deubiquitination	colon cancer lung cancer pancreatic cancer	suppresses activation of CTLs	[96,97]
USP9X	stabilizes PD-L1 through interaction and deubiquitination	oral cancer	inhibits T-cell cytotoxicity	[99]
USP18	deconjugates the ubiquitin-like protein ISG15 from PD-L1	bladder cancer	promotes immune evasion	[100,104]
USP20	stabilizes PD-L1 through deubiquitination.	breast cancer	USP20, stabilized by TINCR, contributes to breast cancer progression	[107]
USP21	stabilizes PD-L1 through interaction and deubiquitination	lung cancer	promotes Treg-cell function	[110,112]
USP22	stabilizes PD-L1 through interaction and deubiquitination	lung cancer liver cancer	inhibits T-cell cytotoxicity	[116,117]

DUBs, deubiquitinating enzymes; OUTB1, OUT domain-containing ubiquitin aldehyde-binding protein 1; PD-L1, programmed death ligand 1; CD8, cluster of differentiation 8; CSN5, constitutive photomorphogenesis 9 (COP9) signalosome 5; USP, ubiquitin-specific protease; NSCLC, non-small-cell lung cancer; IFN, interferon; CTL, cytotoxic T lymphocyte; ISG15, interferon-stimulated gene 15; TINCR, tissue differentiation-inducing non-protein coding RNA; Treg, regulatory T.

## 5. Conclusions and Perspectives

The exploration of PD-L1 regulatory mechanisms, specifically through ubiquitination and deubiquitination, has opened new avenues for understanding cancer immunotherapy. These PTMs are not merely minor biochemical alterations but pivotal factors that dictate the stability and functionality of PD-L1, influencing the immune escape of cancer cells. As discussed in this review, the complexity of these regulatory pathways underscores the intricate interplay between immune surveillance and tumor evasion.

In the context of clinical applications, the insights gained regarding PD-L1 regulation are of paramount importance. They not only refine our understanding of the immune response in cancer but also pave the way for the development of more targeted and effective therapeutic strategies. This is particularly crucial as we witness an expanding horizon of PD-L1/PD-1 inhibitors in cancer treatment. Our review suggests that modulation of PD-L1 through these PTMs could potentially enhance the efficacy of current therapies, offering hope for improved patient outcomes.

Considering the critical roles of E3 ubiquitin ligases and DUBs, researchers are exploring whether targeting these enzymes is an effective strategy for managing PD-L1/PD-1 levels. Shikonin and pyridoxal, as briefly noted in this review, play pivotal roles in the immune response against tumor cells; Shikonin targets CSN5 to accelerate the degradation of PD-L1, while pyridoxal, a form of vitamin B6, enhances the interaction between STUB1 and PD-L1 (Table 3) [56,80]. This results in improved breakdown of PD-L1 and an increase in T-cell-killing activity. Additionally, the emergence of novel therapeutic strategies, including small-molecule inhibitors that focus on blocking PD-L1 deubiquitination, particularly those targeting USPs [91,94,97,117], and the use of proteolysis-targeting chimeras (PROTACs) and the development of antibody-based PROTACs (AbTACs) designed to control PD-L1 ubiquitination [118,119,120,121], marks a significant transformation in cancer therapy approaches (Table 3). These strategies could offer a dual advantage: directly modulating PD-L1 levels and function and potentially circumventing resistance mechanisms that have been a significant hurdle in cancer immunotherapy. This field is poised for deeper exploration of the mechanistic intricacies of PD-L1 regulation. It is imperative that future research delves into the specific pathways and molecular players involved in PD-L1 ubiquitination and deubiquitination in various cancer types. Such a detailed understanding will not only enhance our fundamental knowledge but also guide the development of more precise and effective therapeutic interventions.

Identification of reliable biomarkers related to PD-L1 PTMs is another promising approach. These biomarkers could revolutionize the approach to cancer immunotherapy, allowing for more personalized treatment plans and the ability to predict patient responses to the PD-L1/PD-1 blockade more accurately. Moreover, the exploration of combination therapies integrating PD-L1/PD-1 inhibitors with other immunomodulatory agents is an exciting prospect. Such combinations could potentially overcome the resistance mechanisms that limit the success of monotherapies, thereby significantly improving the clinical outcomes in cancer patients.

Furthermore, while some PTMs may indeed be universal across different organ contexts, others could be organ-specific, reflecting the unique microenvironment and immune landscape of each organ. This variability could influence the efficacy of immune checkpoint inhibitors and may partially explain the variability in response among different cancers or cancer subsets, even when similar biomarkers, such as PD-L1 expression or tumor mutational burden (TMB), are used for patient selection. Understanding these organ-specific differences in PD-L1 PTMs is critical for refining therapeutic targets and strategies, potentially leading to more customized and effective cancer treatments. Finally, the implications of PD-L1 PTMs extend beyond oncology. Investigating their roles in other diseases could reveal new therapeutic potential for immunotherapy, significantly broadening the scope of this field.

In conclusion, the intricate dance of ubiquitination and deubiquitination in regulating PD-L1 presents fertile ground for future research and therapeutic innovation in cancer immunotherapy. As we continue to unravel these complex molecular interactions, we are closer to realizing the full potential of immune-based therapies for cancer treatment.

## Figures and Tables

**Figure 1 ijms-25-02939-f001:**
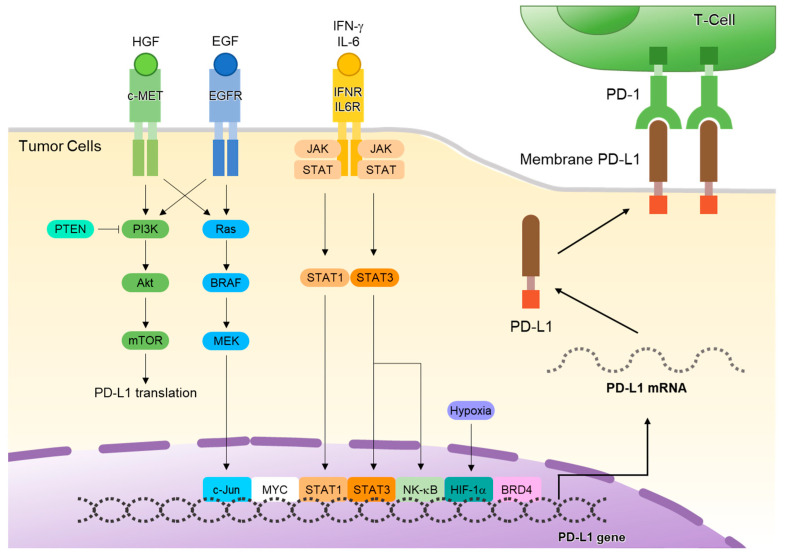
Intracellular signaling pathways regulating programmed death-ligand 1 (PD-L1) expression. Tumor cells evade the antitumor effects of T cells by increasing PD-L1 expression at the transcriptional level by activating different upstream signaling pathways. The signaling pathways, such as JAK/STAT, Ras/MAPK, and PI3K/AKT, play a role in regulating PD-L1 expression primarily by activating key transcription factors. These transcription factors, including STAT3, STAT1, c-Jun, HIF-1α, and NF-κB, move into the nucleus, attach to specific regions on the PD-L1 gene promoter, and stimulate its expression. Abbreviations: HGF, hepatocyte growth factor; EGF, epidermal growth factor; IFN-γ, interferon-gamma; IL-6, interleukin 6; c-MET, cellular–mesenchymal–epithelial transition; PTEN, phosphatase and TENsin homolog; PI3K, phosphoinositide 3 kinase; Akt, Ak strain transforming; mTOR, mammalian target of rapamycin; Ras, rat sarcoma; BRAF, v-Raf murine sarcoma viral oncogene homolog B1; MEK, mitogen-activated protein kinase; IFNR, interferon receptor; IL6R, interleukin 6 receptor; JAK, Janus kinase; STAT, signal transducers and activators of transcription; NF-κB, nuclear factor kappa B; HIF-1α, hypoxia-inducible factor 1-alpha; BRD4, bromodomain-containing protein 4.

**Figure 2 ijms-25-02939-f002:**
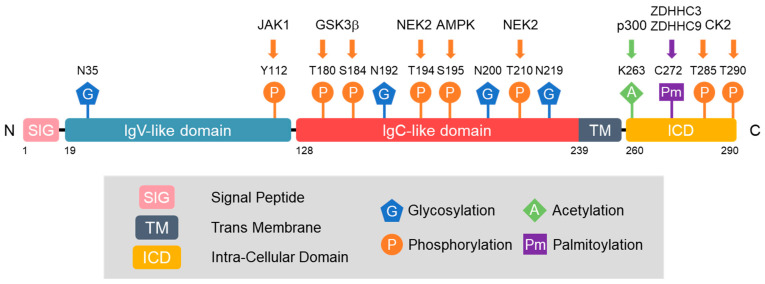
Post-translational modification of PD-L1. The major sites for PD-L1 glycosylation, phosphorylation, acetylation, and palmitoylation are plotted. The enzymes responsible for each type of modification are shown on the upper side. Abbreviations: JAK1, Janus kinase 1; GSK3β, glycogen synthase kinase 3β; NEK2, never in mitosis gene A-related kinase 2; AMPK, adenosine monophosphate-activated protein kinase; p300, p300 acetyltransferase; ZDHHC3, zinc finger Asp-His-His-Cys (DHHC)-type palmitoyltransferase 3; ZDHHC9, zinc finger DHHC-type palmitoyltransferase 9; CK2, casein kinase 2.

**Figure 3 ijms-25-02939-f003:**
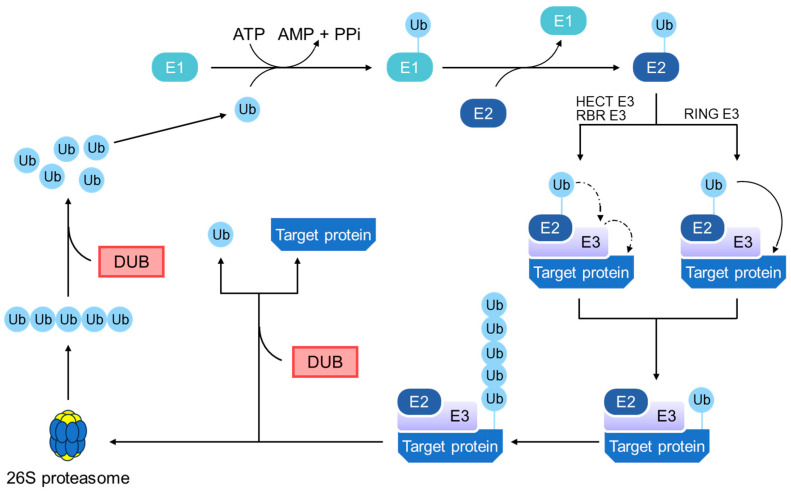
Ubiquitin–proteasome pathway. Ubiquitination is a complex, multistep enzyme cascade crucial for protein degradation within the cell. This process begins with the activation of free ubiquitin (Ub) by an E1 ubiquitin-activating enzyme, a reaction that requires ATP. Following activation, ubiquitin is transferred to an E2 ubiquitin-conjugating enzyme. Subsequently, an E3 ubiquitin-ligase enzyme, which specifically identifies substrate proteins, facilitates the attachment of ubiquitin to the substrate. There are different types of E3 ligases: RING E3 ligases catalyze the direct transfer of Ub from E2 enzymes to the substrate, while HECT and RBR E3 ligases have a catalytic cysteine that intermediates the transfer of Ub from E2 to the target protein. After ubiquitination, the substrate protein is recognized and degraded by the 26S proteasome into small peptides and amino acids. Additionally, the ubiquitination process can be reversed by DUBs, which remove polyubiquitin chains from proteins, thus regulating intracellular ubiquitin levels and ensuring the recycling of ubiquitin for future use. Abbreviations: RING, really interesting new gene; HECT, homologous to E6AP C-terminus; RBR, RING between RING.

**Table 3 ijms-25-02939-t003:** Preclinical and clinical applications targeting ubiquitination of PD-L1.

Drug/Inhibitor	Major Target	Types of Cancer	Action Mechanism	References
Pyridoxal	STUB1		Promotes PD-L1 degradation	[56]
Curcumin	CSN5	Breast cancer	Promotes ubiquitination and degradation of PD-L1	[79]
Shikonin	CSN5	Pancreatic cancer	Inhibits CSN5 and promotes PD-L1 degradation	[80]
Almac4	USP7	Gastric cancer	Downregulates PD-L1 expression and sensitizes cancer cells to T-cell killing	[91]
DUB-IN-2	USP8	Pancreatic cancer	Downregulates PD-L1 expression and activates antitumor immunity	[94]
WP1130	USP9x	Oral squamous cell carcinoma	Inhibits deubiquitination of PD-L1 and destabilizes its expression	[97]
PROTAC BMS-37-C3			Promotes PD-L1 degradation and enhances the killing ability of T cells	[120]
AbTAC AC-1			Induces degradation of PD-L1	[121]

STUB1, stress-inducible protein-1 (STIP1) homology and U-box containing protein 1; PD-L1, programmed death ligand 1; CSN5, constitutive photomorphogenesis 9 (COP9) signalosome 5; USP, ubiquitin-specific protease; DUB, deubiquitinating enzyme; PROTAC, proteolysis-targeting chimera; AbTAC, antibody-based PROTAC.

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
