# Peer review of "Programmed Death Ligand 1 Regulatory Crosstalk with Ubiquitination and Deubiquitination: Implications in Cancer Immunotherapy"

_ijms, 2024, doi:10.3390/ijms25052939_

Round 1
Reviewer 1 Report
Comments and Suggestions for Authors
The review is well written and describes in a very exhaustive manner the role of the post-translational modifications involved in the PD-1 and PD-L1 pathway.
Figures and tables show in a clear manner the role of each post-translational modification in enhancing cancer cell ability to impair immune system through the axis PD-1/PD-L1.
However, the review does not report how clinically these modifications can be targeted and if there is a benefit for specific cancer patients from drugs targeting post-translational modifications involved in the PD-L1 expression or conformation on cancer cells.
I suggest to authors to include a paragraph describing in vivo preclinical studies and clinical trials targeting the post-translational modifications involved in the PD-1 and PD-L1 pathway.
A summary table of these studies would be also helpful for the reader.
Author Response
Response Letter to Reviewers’ comments
Please find the response letter to each comment of the reviewers in a point-by-point manner. We are pleased to resubmit the revised manuscript to International Journal of Molecular Sciences.
Response to Reviewer 1 Comments
The review is well written and describes in a very exhaustive manner the role of the post-translational modifications involved in the PD-1 and PD-L1 pathway. Figures and tables show in a clear manner the role of each post-translational modification in enhancing cancer cell ability to impair immune system through the axis PD-1/PD-L1.
Response: Thank you for your encouraging comments on our review.
However, the review does not report how clinically these modifications can be targeted and if there is a benefit for specific cancer patients from drugs targeting post-translational modifications involved in the PD-L1 expression or conformation on cancer cells. I suggest to authors to include a paragraph describing in vivo preclinical studies and clinical trials targeting the post-translational modifications involved in the PD-1 and PD-L1 pathway. A summary table of these studies would be also helpful for the reader.
Response: Thank you for your constructive feedback. Our review already mentioned some preclinical applications targeting PD-L1 ubiquitination in the section 4. Ubiquitination and Deubiquitination of PD-L1. Additionally, based on your suggestion, we have compiled "Table 3. Preclinical and clinical applications targeting ubiquitination of PD-L1," which summarizes these studies, offering readers a clear overview of current advances and the clinical relevance of targeting PD-L1 PTMs in cancer therapy.

Reviewer 2 Report
Comments and Suggestions for Authors
This review article focuses on the regulation of ubiquitination and deubiquitination of PD-L1. However, similar articles have been published, such as Genes & Diseases Volume 10, Issue 3, May 2023, Pages 848-863. and Mol Ther. 2021 Mar 3; 29(3): 908–919. Some content is similar to this article, so the innovation is insufficient. Moreover, many parts of the content written in the article are repeated, such as lines 37-38 and 94-95, lines 163-168 and 207-215. In addition, line 102 mentions ''interferon-gamma, which is secreted by tumor-infiltrating lymphocytes and cancer cells within the tumor,'' but Figure 1 and Reference 19 shows that TIL secretes interferon-gamma. Please revise it.
Author Response
Response Letter to Reviewers’ comments
Please find the response letter to each comment of the reviewers in a point-by-point manner. We are pleased to resubmit the revised manuscript to International Journal of Molecular Sciences.
Response to Reviewer 2 Comments
This review article focuses on the regulation of ubiquitination and deubiquitination of PD-L1. However, similar articles have been published, such as Genes & Diseases Volume 10, Issue 3, May 2023, Pages 848-863. and Mol Ther. 2021 Mar 3; 29(3): 908–919. Some content is similar to this article, so the innovation is insufficient.
Response: We acknowledge the reviewer's concern regarding the perceived similarity of our manuscript to previously published articles. Our review expands the discussion by introducing additional E3 ubiquitin ligases and deubiquitinating enzymes not covered in the previously mentioned articles. We believe these additions and clarifications significantly enhance the manuscript's contribution to the current understanding of PD-L1's regulatory mechanisms in the context of cancer immunotherapy.
Moreover, many parts of the content written in the article are repeated, such as lines 37-38 and 94-95, lines 163-168 and 207-215.
Response: We appreciate the reviewer pointing out the specific lines where content repetition occurred. We have carefully revised these sections to eliminate redundancy and ensure a concise and coherent flow of information throughout the manuscript. The changes aim to improve readability and maintain the reader's engagement.
In addition, line 102 mentions ''interferon-gamma, which is secreted by tumor-infiltrating lymphocytes and cancer cells within the tumor,'' but Figure 1 and Reference 19 shows that TIL secretes interferon-gamma. Please revise it.
Response: Thank you for highlighting the inconsistency regarding the source of interferon-gamma as mentioned in line 102, Figure 1, and Reference 19. we have amended the figure for clarity.

Reviewer 3 Report
Comments and Suggestions for Authors
It is a very well organized review on the PD-L1 and post translational modification. For the readers
1. Various kind of modifications are stated and are they possibly different depending on organ context? Address you expect using these modification system as therapeutic targets (probably combined with current ICI therapy, right?) , give some insights on organ specific (or these are generally the same) difference of PTM of PD-L1. ICIs are effective some cancers or some subset of cancers, not the other cancers even though similar criterial such as IHC of PD-L1 or TMB are used. Does this reflect the difference in PTM difference?
2. Some (not many) cases have genomic amplification of PD-L1 (0.5% of NSCLC) and they are excellent response to ICI (Inoue et al. JAMA network). These are excessive amount of protein which override PTM?
Author Response
Response Letter to Reviewers’ comments
Please find the response letter to each comment of the reviewers in a point-by-point manner. We are pleased to resubmit the revised manuscript to International Journal of Molecular Sciences.
Response to Reviewer 3 Comments
It is a very well-organized review on the PD-L1 and post translational modification. For the readers
- Various kind of modifications are stated and are they possibly different depending on organ context? Address you expect using these modification system as therapeutic targets (probably combined with current ICI therapy, right?), give some insights on organ specific (or these are generally the same) difference of PTM of PD-L1. ICIs are effective some cancers or some subset of cancers, not the other cancers even though similar criterial such as IHC of PD-L1 or TMB are used. Does this reflect the difference in PTM difference?
Response: we have expanded our discussion to address the possibility of organ-specific differences in PD-L1 PTMs and their implications for therapeutic strategies. We propose that while some PTMs may indeed be universal across different organ contexts, others could be organ-specific, reflecting the unique microenvironment and immune landscape of each organ. This could influence the efficacy of ICIs and may partially explain the variability in response among different cancers or cancer subsets, even when similar biomarkers, such as PD-L1 expression or tumor mutational burden (TMB), are used for patient selection.
- Some (not many) cases have genomic amplification of PD-L1 (0.5% of NSCLC) and they are excellent response to ICI (Inoue et al. JAMA network). These are excessive amount of protein which override PTM?
Response: The study by Inoue et al., published in JAMA Network Open in 2020, concludes that PD-L1 amplification is associated with a favorable response to nivolumab monotherapy among NSCLC patients, suggesting that genomic amplification can indeed have a significant impact on treatment outcomes, seems to override the subtler effects of PTMs on PD-L1 function.
High PD-L1 expression, whether due to genomic amplification or other mechanisms, generally suggests a favorable response to ICI therapy. Meanwhile, the impact of PTMs on ICI treatment efficacy may depend on the specific modifications and how they affect PD-L1 function and interaction with immune cells. Although the specific roles of PTMs were not directly addressed in this study, it is well-known that PTMs can significantly affect protein stability, degradation, localization and interaction with PD-1.
Both PD-L1 expression levels and PTMs play significant roles in the regulation of immune responses in cancer. However, their relative importance can vary based on the context of the tumor microenvironment and individual patient characteristics.
These modifications can modulate the immunosuppressive function of PD-L1 and potentially affect the response to ICI therapy. Therefore, further research would be needed to fully understand this relationship.

Reviewer 4 Report
Comments and Suggestions for Authors
The review titled: PD-L1 Regulatory Crosstalk with Ubiquitination and Deubiquitination: Implications in Cancer Immunotherapy was submitted by Kim et al. to the International Journal of Molecular Sciences. This manuscript is timely, of potential therapeutic importance and appropriate for this journal.
In the manuscript, Kim et al. discuss the immune checkpoint PD-1 and its ligand PD-L1 and the ever increasing importance of immune checkpoint inhibitors in cancer immunotherapy. The regulatory pathway involving ubiquitination and deubiquitination is well summed up. The involvement of E3 ubiquitin ligases in PD-L1 regulation and their potential in improving the efficacy of immunotherapies and potential use as targets for immunotherapy are discussed. On the other hand, the role of deubiquitinating enzymes in stabilizing PD-L1 expression, thereby increasing the immunosuppressive function of this immune checkpoint, is described. The use of these enzymes as potential therapeutic targets is eluded to. The authors conclude that the development of more targeted and effective therapeutic strategies could lead to improved patient outcomes.
Kim et al. suggest that future research should focus on specific pathways in PD-L1 ubiquitination and deubiquitination in various types of cancer. In addition, the identification of reliable biomarkers would allow for more personalized treatment plans.
It should be noted that much of the content in this manuscript has been covered recently in a review by Ding et al. (Ding, P.; Ma, Z.; Fan, Y.; Feng, Y.; Shao, C.; Pan, M.; Zhang, Y.; Huang, D.; Han, J.; Hu, Y.; Yan, X. Emerging role of ubiquitination/deubiquitination modification of PD-1/PD-L1 in cancer immunotherapy. Genes Dis 2023, 10, (3), 848-863. doi: 10.1016/j.gendis.2022.01.002). The present review does, however, build on the knowledge base in the review by Ding et al., particularly in the section on post-translational modification (PTMs) of PD-L1. Additional E3 ubiquitin ligases and deubiquitinating enzymes are discussed.
The authors have provided a thorough and focussed overview of the literature on the topic with recent literature being cited. The review follows a logical sequence. The use of English in the manuscript is excellent and appropriate scientific language has been used.
Author Response
Response Letter to Reviewers’ comments
Please find the response letter to each comment of the reviewers in a point-by-point manner. We are pleased to resubmit the revised manuscript to International Journal of Molecular Sciences.
Response to Reviewer 4 Comments
The review titled: PD-L1 Regulatory Crosstalk with Ubiquitination and Deubiquitination: Implications in Cancer Immunotherapy was submitted by Kim et al. to the International Journal of Molecular Sciences. This manuscript is timely, of potential therapeutic importance and appropriate for this journal.
In the manuscript, Kim et al. discuss the immune checkpoint PD-1 and its ligand PD-L1 and the ever increasing importance of immune checkpoint inhibitors in cancer immunotherapy. The regulatory pathway involving ubiquitination and deubiquitination is well summed up. The involvement of E3 ubiquitin ligases in PD-L1 regulation and their potential in improving the efficacy of immunotherapies and potential use as targets for immunotherapy are discussed. On the other hand, the role of deubiquitinating enzymes in stabilizing PD-L1 expression, thereby increasing the immunosuppressive function of this immune checkpoint, is described. The use of these enzymes as potential therapeutic targets is eluded to. The authors conclude that the development of more targeted and effective therapeutic strategies could lead to improved patient outcomes.
Kim et al. suggest that future research should focus on specific pathways in PD-L1 ubiquitination and deubiquitination in various types of cancer. In addition, the identification of reliable biomarkers would allow for more personalized treatment plans.
It should be noted that much of the content in this manuscript has been covered recently in a review by Ding et al. (Ding, P.; Ma, Z.; Fan, Y.; Feng, Y.; Shao, C.; Pan, M.; Zhang, Y.; Huang, D.; Han, J.; Hu, Y.; Yan, X. Emerging role of ubiquitination/deubiquitination modification of PD-1/PD-L1 in cancer immunotherapy. Genes Dis 2023, 10, (3), 848-863. doi: 10.1016/j.gendis.2022.01.002). The present review does, however, build on the knowledge base in the review by Ding et al., particularly in the section on post-translational modification (PTMs) of PD-L1. Additional E3 ubiquitin ligases and deubiquitinating enzymes are discussed.
The authors have provided a thorough and focused overview of the literature on the topic with recent literature being cited. The review follows a logical sequence. The use of English in the manuscript is excellent and appropriate scientific language has been used.
Response: We sincerely appreciate the constructive and encouraging comments provided by the reviewer regarding our manuscript. Your positive feedback is immensely valuable to us, and we are grateful for the recognition of our work's potential contribution to the field.
